# Competition Seriousness and Competition Level Modulate Testosterone and Cortisol Responses in Soccer Players

**DOI:** 10.3390/ijerph17010350

**Published:** 2020-01-04

**Authors:** Manuel Jiménez, José Ramón Alvero-Cruz, Juan Solla, Jorge García-Bastida, Virginia García-Coll, Iván Rivilla, Enrique Ruiz, Jerónimo García-Romero, Elvis A. Carnero, Vicente Javier Clemente-Suárez

**Affiliations:** 1Facultad de Educación, Universidad Internacional de La Rioja, Avenida de la Paz 137, 26002 Logroño, Spain; jorge.garcia.bastida@unir.net (J.G.-B.); virginia.garciacoll@unir.net (V.G.-C.); ivan.rivilla@unir.net (I.R.); enriquejruiz@yahoo.es (E.R.); 2Facultad de Medicina, Universidad de Málaga, Campus Teatinos s/n, 29071 Málaga, Spain; alvero@uma.es (J.R.A.-C.); jeronimo@uma.es (J.G.-R.); 3Grupo de Investigación Hi20, Universidad de Vigo, Campus a Xunqueira s/n, 36005 Pontevedra, Spain; 4Research Institute for Diabetes and Metabolism, Florida Hospital Sanford, Burnham Prebys Medical Discovery Institute, 301 East Princeton Street, Orlando, FL 32804, USA; Elvis.Carnero@flhosp.org; 5Faculty of Sports Science, Universidad Europea de Madrid, 28670 Madrid, Spain; vctxente@yahoo.es; 6Grupo de Investigación en Cultura, Educación y Sociedad, Universidad de la Costa, Barranquilla 080002, Colombia

**Keywords:** soccer, competitive behaviour, winner effect, social dominance, testosterone, cortisol

## Abstract

This study aimed to analyze the modulating effect of competition seriousness and competition level in the testosterone and cortisol responses in professional soccer player. Ninety five (95) soccer players were included in this study (professional, *n* = 39; semiprofessional, *n* = 27; amateur, *n* = 29) before and after training, friendly game and official games. Repeated measures ANOVA showed higher testosterone levels (F_(1,89)_ = 134, *p* < 0.0001, η^2^_p_ = 0.75) in professional soccer players, when compared with semiprofessional (*p* < 0.0001) or amateur athletes (*p* < 0.0001). After winning a competition game an increase in testosterone levels was observed in professionals (t = −3.456, *p* < 0.001), semiprofessionals (t = −4.400, *p* < 0.0001), and amateurs (t = −2.835, *p* < 0.009). In contrast, this momentary hormonal fluctuation was not observed after winning a friendly game or during a regular training day. Additionally, statistical analysis indicated that cortisol levels were lower in professional (t = −3.456, *p* < 0.001) and semiprofessional athletes (t = −4.400, *p* < 0.0001) than in amateurs (t = −2.835, *p* < 0.009). In soccer players a rise in testosterone was only observable when the team was faced with an actual challenge but did not support a different response between categories. Thus, the desire to achieve a goal (and keep the social status) may be one of the key reasons why testosterone levels rise promptly. Conversely, testosterone did not change after friendly games, which suggests these situations are not real goals and the players do not perceive an actual threat (in terms of dominance) more than the preparation for their next competitive game.

## 1. Introduction

An optimal performance in soccer is determined by a multifactorial coordination of physiological, psychological and sociological variables [1]. Although the research of physiological and biological determinants has received traditionally more attention, the relationship between environmental variables (e.g., performance demands, social status, etc.) could be an interesting research aim. Thus, hormones such us testosterone and cortisol have been related with the ability to perform soccer skills at high speed and recovery capacity [2,3,4,5], and their release is driven not only by internal triggers but also regulated by environmental stimulus through different areas at central nervous system [6,7,8,9]. In this line, it has been widely reported that the capability to modulate humoral factors was related with success in sport games competitions [10,11,12].

Professional soccer games are viewed by more than 100 million people around the world, which must be external factor affecting humoral status of soccer players. So, competitive duels outcome may be influenced by the neuroendocrine response patterns that can change after victory or defeat [13]. Some dominant behaviours that motivate and modulate the human being to compete against each other, or to be aversive rivals are related to neuroendocrine responses [10,11,12,13]. According to several authors, high testosterone concentrations (T) have been related to social status, seeking and dominating behaviours in animal and human studies [6,14]. Mazur’s bio-social model explained how circulating T levels rise or fall as a consequence of outcomes in competitive events and fighting for incentives [15], but the “winner effect” is not just related to T elevation in winners, also that dominant personalities could show higher basal T concentrations than subordinates [16]. Then, high T concentrations may be related to a high social status and dominating behaviours since high T is presumed to support the expression of high-status signs, while low T shifts signalling toward deferent signs [15]. However, testosterone is not the only hormone implicated in competitive behaviour. Cortisol concentrations (C) have also been described as a determinant of physical activation by preparing body to fight and like and excellent psychosocial stress biomarker [11,13]. The hormonal stress response is modulated by different contextual factors that would induce different responses depending on the psychological interpretation of the subject of the environment. Factors as motivation, the seriousness of the competition and competition level are main parameters that would modulate the hormonal stress response, being these parameters, poor join studied [17].

To improve knowledge in this new and interesting area we proposed the present research with the aim of to analyse the modulating effect of competition seriousness and competition level in the testosterone and cortisol responses in professional soccer player.

The initial hypothesis was that the type of contest (i.e., friendly vs. official games) would affect the hormone releases on the different competition level (amateur, semiprofessional and professional categories); in this scenario, the type of competition category was an indicator of status position in Spanish Soccer Leagues.

## 2. Material and Methods

### 2.1. Participants

A total of ninety-five soccer players from three different categories of Spanish soccer leagues participated in this study. All players made up part of the top-8 teams in their respective soccer leagues. A medical screening was carried out to exclude endocrine and psychiatry disorders, and drug consumption (i.e., drugs or medicaments). All players passed dope controls during the regular seasons in their respective leagues. All participants met at least three weekly training days lasting between 100 to 120 min. All volunteers who met these previous inclusion criteria and signed the informed consent of the study were accepted as participants. This protocol and all its procedures were approved by the Ethical Committee of University of Malaga, Malaga, Spain with number 26-2018-H in accordance with the Declaration of Helsinki for medical studies with humans.

### 2.2. Procedure

Participants provided two saliva samples (2–4 mL) 40 min before ((PRE) always before the warm-up phase) and 40 min after (POST) training, friendly or competition games. The average time between PRE and POST samples was three hours (no food, brushing teeth, protein intakes were used on this time) and all games were played between 10:00 a.m. and 2:00 p.m. Fluctuations produced by time collection, circadian cycles or warm-up influences were adequately reduced in this protocol. Participants could not brush their teeth, drink or eat in the previous 30 min prior to saliva collection. Whole saliva samples were collected in Salivette^®^ swab tubes (Sarstedt, Nümbrecht, Germany), then they were frozen at −40 °C. Samples were 15 min centrifuged at 3000 rpm to obtain about 2 mL of whole saliva supernatant. Cortisol and testosterone were analysed from the saliva specimens in duplicate by ELISA immunoassay kits (Diametra^®^, Milan, Italy) with a Triturus^®^ multi-test immunoassay system (Grifols, Barcelona, Spain). Intra and inter-assay coefficients of variation were 5.7 and 12.6% for T, and 5.2 and 11.9% for C (minimum detection limit 3.8 pg/mL y 0.5 ng/mL for T and C), respectively.

The study was conducted between July and December 2016 and all friendly and competition games ended in victory for the home team. No other physiological variables were assessed due to Fédération Internationale de Football Association (FIFA) rules, so physical activity influence over T or C fluctuations could not be assessed. Hormonal samples were taken in the last 2 months of the 2016/2017 season, and during this period professional players were involved in play-offs to qualify for European competitions (UEFA, Euro-league cup) or to move into Liga Santander, and semiprofessional and amateur players were involved in play-off matches to reach professional and semiprofessional leagues, respectively. All players were paid to play with average annual wages of 800,000, 80,000 y 15,000 € for the three categories respectively; also, the final classification in the league had repercussions for the following season in sponsorship and revenues from TV rights. As all players made money playing, categories were called “professional”, “semiprofessional” or “amateur” depending of Spanish Football Federation’s official nomenclatures. Accordingly, official games helped them to reach or maintain their socioeconomic status, category and the social impact they provide to their teams’ supporters. We speculate that these circumstances, and incentives are the major reasons why the players perceive official games as a much more serious situation compared to friendly games, where there are no social status threats or tangible incentives to earn. Consequently, this approach could be a valid construct to prove or refute our initial hypothesis about seriousness of competition as a modulator of testosterone response patterns.

### 2.3. Statistical Analysis

All statistics were performed using the IBM SPSS 20 package (IBM Corp, Armonk, NY, USA). Repeated measures analysis of variance (2 × 3 × 3 ANOVA) was carried out to compare within (PRE and POST) and between-subjects (3 competition categories and 3 game types) differences. Multiple comparisons between categories and game type were performed by using Tukey’s *post hoc* tests. Additionally, paired-sample T-tests were used to analyse differences POST-PRE for each individual group. Differences were analysed as relative percentage change in hormone levels [(post-competition minus pre-competition)/(pre-competition) × 100]; following prior studies [10,11]. Effect sizes were obtained by η^2^_p_. The homogeneity of the variances of the dependent variables was assessed by Kolmogorov-Smirnov tests, showing that testosterone and cortisol were not normally distributed; the hormonal variables were logarithmically transformed to perform parametric analysis (non-transformed data is shown in the table to facilitate comparison with prior studies). To analyse type II error 1-β was calculated as well. For all procedures, a level of *p* ≤ 0.05 was selected to indicate statistical significance.

## 3. Results

Participants characteristics: thirty-nine were professionals (26.76 ± 4.05 years; body mass index 22.77 ± 1.16 kg/m^2^; category First and Second Division of the Spanish Professional BBVA League), twenty-seven semiprofessionals (25.07 ± 2.92 years; body mass index 23.05 ± 0.96 kg/m^2^; category Second Division B of the National Soccer League) and twenty-nine amateurs (25.01 ± 3.31 years; body mass index 22.96 ± 1.05 kg/m^2^; category Third Division of the National League).

Testosterone and cortisol values for all competition and game categories are shown in Table 1. These results revealed that T was significantly different depending upon the competition category (F_(1.89)_ = 134,01 *p* < 0.0001, η^2^_p_ = 0.75); specifically, professional players had higher hormone concentrations than semiprofessionals and amateurs (*p* < 0.0001 for both, Table 1. Regarding C concentrations, we also observed differences among competition categories (F_(1.89)_ = 9.53, *p* < 0.0001, η^2^_p_ = 0.18); so, C was significantly lower in professionals and semiprofessionals than amateur players (*p* < 0.0001 and *p* < 0.021, respectively). In addition, we found an interaction effect between C and game type (F_(1.89)_ = 4.811, *p* = 0.031, η^2^_p_ = 0.05; see differences in Table 1.

In Figure 1 we provide the results of % change in T across competition level and game type (differences between friendly and competitive games); data support the relevance of the seriousness of the game with independence of competition level: professionals (t = −3.456, *p* < 0.001), semiprofessionals (t = −4.400, *p* < 0.0001), and amateurs (t = −2.835, *p* < 0.009). There were no differences between training and friendly games. Post hoc 1- was over 0.90 on ANOVA and t-Student results and critical F and t values were clearly surpassed.

## 4. Discussion

This research aimed to analyse the modulating effect of competition seriousness and competition level in T and C responses in professional soccer player. In this study, we provide evidence for a higher testosterone production in professional than in non-professional categories, which is affected by the type of competition. This result confirms seriousness of the competition (i.e., friendly vs. official games) as modulators of hormonal response patterns. Sports encounters lead to moods of euphoria or frustration relative to the outcome of the event (as argued, [11]), and provide an excellent opportunity to study the social T-dominance relationship paradigm after winning an official game [6,13,15].

Our results showed that soccer players of the highest performance level and category (professionals) produced the highest level of T and the lowest C; conversely, amateur athletes showed the lowest T and highest C. Moreover, the change in the androgen hormone (POST-PRE) was dependent on the seriousness of the game (i.e., official versus practice, training, etc.); thereby illustrating that the changes are only significant in competition games when an actual threat to social status was plausible. This data may be consistent with the Biosocial Model theory and linked with the T-Dominance paradigm [14,18,19]. Likewise, these findings agree with a previously reported study suggesting that in competition games there is much more at stake to be won and lost in terms of dominancy than in friendly or training matches [20].

Regarding testosterone differences among the various categories of competition level, our data support a direct relationship between status and higher androgen production [21]. In this line, our sample of professional and semiprofessional athletes had a significantly higher income than amateurs and the average Spanish income, which may confer a higher level of socioeconomic status, therefore we speculate the significant differences in T concentrations between professional/ semiprofessional and amateur players could be related with different social status and higher access to resources [15,18,22]. Furthermore, in human dominance research, T has been positively associated with victory in sport events [11,16], hence, it is more likely for usual winners to reach professional status with high level teams, sign higher monetary contracts and have a greater impact on mass media or team supporters. In our sample, the higher category and greater social status were related to higher T concentrations in the three types of situations investigated: official game, friendly game and training day.

Cortisol responses were also category-dependent, and these differences between competition levels could be explained in two ways, physiological adaptation and familiarization with stressful events or situations. Firstly, the high volumes of training that are part of the professional soccer players and semiprofessionals life (typically, they work out 6–7 days a week) have been related to a lower reactivity to psychosocial stressors [23]. Also, professional players presented low C concentrations than non-professionals throughout the season [24]. A lower C production may be better physically tolerated and results in a reduction of the allostatic load produced by high concentrations of circulating glucocorticoids maintained over time, offering an advantage to reach and maintain a higher social rank [21]. This rationale was consistent with our results showing the highest peak of C in amateur players after a competitive game and before friendly and training events. Secondly, negative emotional valences and psychological stressors (e.g., defeat frustration) increased circulating C in laboratory studies and in sports competition [11,13,25]. Low C concentrations would aid better performance, not just in physiological efforts, but also in psychological threats and social status seeking [21,26]. Hence, better regulation of cortisol production on experienced players could help in their challenge appraisals abilities [27,28]. Also, in extreme sports, reduced Hypothalamic-Pituitary-Adrenal axis activity has been observed in repetitive exposure to stressful situations [29]. It is tempting to speculate that low or moderate C concentrations as a neuroendocrine condition to becoming professional athlete in the game of soccer, sport traditionally related with weekly competitive challenges. Modifications in T were observed in all categories of official games, but not in friendly games. Serious games (i.e., official games) were perceived as real-life conditions, because in these situations social status may be threatened, and important incentives could be won or lost. Mazur’s model was demonstrated here with important increases in T prevalent after a win, compared with no differences in T levels after friendly games. This “winner effect” has been studied in lab settings and has shown circulating T increases following consecutive competition days, feeds competitive behaviours and future success [30]. We found how the seriousness of competition is not a category-dependent effect as each soccer player seemed to be motivated to win, and only in the official games could be observed that competition level was able to modulate hormonal response patterns. The seriousness of competition, therefore, could be an important variable for the design of future research in sports competition and could provide a new and better understanding of the relevance of hormonal response patterns and incentives.

## 5. Limitation of the Study

The principal limitation of the present research was that players included belong to the most successful group in their respective categories, so we cannot confirm that testosterone and cortisol levels are not different in players from teams facing more stressful scenarios, for example relegation. T and C basal data on a resting day were not included in this study. Due to constraints related with the rules in the professional leagues, we were not able to collect additional physiological and psychometric information to complete a more mechanistic hypothesis and our results may not imply causality but association.

## 6. Conclusions

In soccer players a rise in testosterone was only observable when the team was faced with an actual challenge but did not support a different response between categories. Thus, the desire to achieve a goal (and keep the social status) may be one of the key reasons why testosterone levels rise promptly. Conversely, testosterone did not change after friendly games, which suggests these situations are not real goals and the players do not perceive an actual threat (in terms of dominance) more than the preparation for their next competitive game. These results endorse the importance of selecting professional athletes during official competitions to understand better the mechanisms underlying the Testosterone-Dominance relationship in humans

## Figures and Tables

**Figure 1 ijerph-17-00350-f001:**
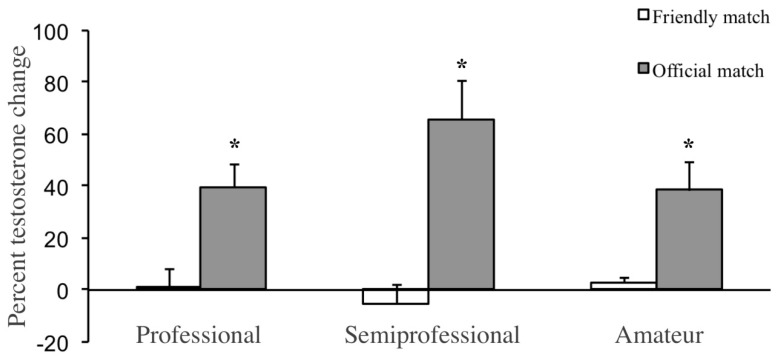
Percent testosterone change in friendly and official games in professional, semiprofessional and amateur soccer players. Statistically significant differences in percent testosterone change (* *p* < 0.01) compared friendly game with official game in professional, semiprofessional and amateur soccer players.

**Table 1 ijerph-17-00350-t001:** Salivary testosterone and cortisol concentrations before and after the training, friendly game and official games in professional (*n* = 39), semiprofessional (*n* = 27) and amateur (*n* = 29) soccer players (mean ± standard error of mean).

	**Training Game**			**Friendly Game**			**Competition Game**		
**Testosterone (pg/mL)**	**Professional**	**Semiprofessional**	**Amateur**	**Professional**	**Semiprofessional**	**Amateur**	**Professional**	**Semiprofessional**	**Amateur**
Before	321.4 ± 18.2 ***	202.4 ± 6.9 ###	126.2 ± 2.4 £££	283.2 ± 33.1	210.4 ± 13.6	125.3 ± 3.0 £££	249.9 ± 14.1	224.2 ± 15.2 ###	113.8 ± 6.8 £££
After	315.3 ± 20.1 ***	203.6 ± 8.6 ###	120.1 ± 2.5 £££	269.4 ± 28.1	190.4 ± 10.8	128.0 ± 3.1 £££	335.8 ± 19.6	355.9 ± 26.9	147.9 ± 7.7 $$
	**Training Game**			**Friendly Game**			**Competition Game**		
**Cortisol (ng/mL)**	**Professional**	**Semiprofessional**	**Amateur**	**Professional**	**Semiprofessional**	**Amateur**	**Professional**	**Semiprofessional**	**Amateur**
Before	6.4 ± 0.6	4.6 ± 0.3 ###	10.3 ± 1.3 £££	5.8 ± 0.9	6.2 ± 1.0	7.5 ± 1.3	5.5 ± 0.8	9.2 ± 0.8	10.2 ± 2.1 £££
After	4.4 ± 0.4	5.3 ± 0.4	6.3 ± 0.7 £££	5.5 ± 1.0	7.0 ± 1.3	9.9 ± 2.0 £££	6.5 ± 0.7	7.9 ± 1.0 #	18.1 ± 3.5 £££

*** *p* < 0.001 for significant differences in testosterone and cortisol concentrations between professional and semiprofessional players; # *p* < 0.05 for differences between semiprofessional and amateur players; ### *p* < 0.001 respectively for differences between semiprofessional and amateur players; £££ *p* < 0.001 for differences between professional and amateur players; $$ *p* < 0.01 for professional and semiprofessional, and amateur.

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
