# Peer review of "Competition Seriousness and Competition Level Modulate Testosterone and Cortisol Responses in Soccer Players"

_ijerph, 2020, doi:10.3390/ijerph17010350_

Round 1
Reviewer 1 Report
DEAR AUTHORS
This study is very interesting and well designed. The paper is nicely written and the topic is very relevant.
General comments
Abstract
In the first sentence, “this study aimed to the modulating effect” I would rephrase to “aimed to analyze/ investigate the modulating effect
The sentence “ 95 soccer players were analyzed” replace for were included in this study.
Key-words: review for MESH terms
Introduction
The introduction is well written and current.
M&M
In this section only data regarding methods should be kept. BMI should be described in the Results section.
“Drug consumption” is referring to illicit drugs? Please clarify in the text.
The term “saliva” is ambivalent, since it doesn’t differentiate between salivary secretions as they exit the ductal systems and whole saliva, which is the mixed fluid accumulating in the oral cavity. In the salivary research domain it is therefore important to differentiate between pure salivary secretions and whole saliva. When the authors mentioned “saliva” I understand they are referring to “Whole saliva”, which represents a mixture of exocrine secretion coming into contact with the oral microbiome, gingival crevicular fluid, a variety of hosts cells and food debris. Whole saliva, therefore, is a turbid suspension since it contains cellular elements. Removal of these cellular components by centrifugation leads to a clear solution known as whole saliva supernatant (WSS), which is amenable to biochemical investigation. Please clarify this in the text.
Neat or diluted saliva supernatant samples were used for the ELISA assay? Neat saliva is viscous while diluted improves the Optical Density (O.D.) for the spectrophotometric reading. Measurements within 1-2 mg/ml total protein amount per sample are advisable for salivary immune based assays. Usually low concentration dilutions are applicable, like 1:2 or 1:5.
Author Response
DEAR AUTHORS
This study is very interesting and well designed. The paper is nicely written and the topic is very relevant.
Thank you
General comments
Abstract
In the first sentence, “this study aimed to the modulating effect” I would rephrase to “aimed to analyze/ investigate the modulating effect
This sentence was changed
The sentence “ 95 soccer players were analyzed” replace for were included in this study.
This sentence was changed
Key-words: review for MESH terms
Key-words were adapted to MESH
Introduction
The introduction is well written and current.
M&M
In this section only data regarding methods should be kept. BMI should be described in the Results section.
Descriptives were written in Results section
“Drug consumption” is referring to illicit drugs? Please clarify in the text.
it was clarified on the manuscript
The term “saliva” is ambivalent, since it doesn’t differentiate between salivary secretions as they exit the ductal systems and whole saliva, which is the mixed fluid accumulating in the oral cavity. In the salivary research domain it is therefore important to differentiate between pure salivary secretions and whole saliva. When the authors mentioned “saliva” I understand they are referring to “Whole saliva”, which represents a mixture of exocrine secretion coming into contact with the oral microbiome, gingival crevicular fluid, a variety of hosts cells and food debris. Whole saliva, therefore, is a turbid suspension since it contains cellular elements. Removal of these cellular components by centrifugation leads to a clear solution known as whole saliva supernatant (WSS), which is amenable to biochemical investigation. Please clarify this in the text.
It was clarified into the text.
Neat or diluted saliva supernatant samples were used for the ELISA assay? Neat saliva is viscous while diluted improves the Optical Density (O.D.) for the spectrophotometric reading. Measurements within 1-2 mg/ml total protein amount per sample are advisable for salivary immune based assays. Usually low concentration dilutions are applicable, like 1:2 or 1:5.
Grifols Triturus is a completely automatized divice, and the automatic procedure homogenize samples into the machine before they were inmunoassayed. This is very interesting to be used in hormonal research.
Reviewer 2 Report
This paper by M. Jimenez et al aims at proving differences in testosterone (T) and cortisol (C) levels in soccer players, before and after games and as a function of players' status and of game importance (for fun, practice or competition).
The goal and rationale of the study are not very clear. Consequently, the conclusions are vague although the title is too affirmative.
Data in the Figure are just calculated from data already in the table. These percentages should rather be included in the table and the figure should be withdrawn.
It is not clear how many times each player was tested. For each value in the table, it would be necessary to indicate n. Is it always the number of players and the same numer of samples from each, in each category and for all types of games ?
The basal values of T and C should have been obtained outside of games, out of any stress due to coming games.
It is not clear why T is significantly lower in Pro players before a competition game relative to other types of games. And it seems it goes back to normal during the game.
Author Response
This paper by M. Jimenez et al aims at proving differences in testosterone (T) and cortisol (C) levels in soccer players, before and after games and as a function of players' status and of game importance (for fun, practice or competition).
The goal and rationale of the study are not very clear. Consequently, the conclusions are vague although the title is too affirmative.
The rationale of this study was explained in last paragraph of the introduction, finalizing this with the research aim. The final conclusion response to research aim proposed in the manuscript.
Data in the Figure are just calculated from data already in the table. These percentages should rather be included in the table and the figure should be withdrawn.
In prior studies, percentual changes were important to understand how hormonal momentary fluctuations are related with some competitive behaviors. For example, Jiménez et al. (2012), showed similar percentual changes to victory and defeat in both genders, authors suggested evidence to same response patterns to competition. We considered relevant this graphic to be compared the effect of seriousness games with friendly games. Table 1 shows total hormonal changes but not the percentual changes.
(Jiménez, M., Aguilar, R., & Alvero-Cruz, J. R. (2012). Effects of victory and defeat on testosterone and cortisol response to competition: evidence for same response patterns in men and women. Psychoneuroendocrinology, 37(9), 1577-1581.)
It is not clear how many times each player was tested. For each value in the table, it would be necessary to indicate n. Is it always the number of players and the same numer of samples from each, in each category and for all types of games ?
On Table 1. legend there was introduced each category n. All players were tested each time, so the number of the players were the same in each moment and category.
The basal values of T and C should have been obtained outside of games, out of any stress due to coming games.
That is correct. But in this study we decided provide an "active" sample (training day) to be compared with other "active" moments (friendly and official games).
It is not clear why T is significantly lower in Pro players before a competition game relative to other types of games. And it seems it goes back to normal during the game.
That was not clear, of course. But this "low or medium" T before games was also observed in many other poor studies:
Jiménez, M., Aguilar, R., & Alvero-Cruz, J. R. (2012). Effects of victory and defeat on testosterone and cortisol response to competition: evidence for same response patterns in men and women. Psychoneuroendocrinology, 37(9), 1577-1581.
Aguilar, R., Jiménez, M., & Alvero-Cruz, J. R. (2013). Testosterone, cortisol and anxiety in elite field hockey players. Physiology & behavior, 119, 38-42.
Mazur, A., & Lamb, T. A. (1980). Testosterone, status, and mood in human males. Hormones and behavior, 14(3), 236-246.
Sometimes, opponet´s psychological state could modulate T and C response.
van der Meij, L., Buunk, A. P., Almela, M., & Salvador, A. (2010). Testosterone responses to competition: the opponent's psychological state makes it challenging. Biological psychology, 84(2), 330-335.
Reviewer 3 Report
The authors have laidout background and necessity for validating an instrument for measuring in ostomates. The methods used for the purpose of the work were appropriate.
The authors also performed temporal stability verification on a small sub-sample for reliability. In addition, the relationship between items also analyzed.
The work is important because having a valid and reliable instrument
Author Response
Comments and Suggestions for Authors
The authors have laidout background and necessity for validating an instrument for measuring in ostomates. The methods used for the purpose of the work were appropriate.
Thank you
The authors also performed temporal stability verification on a small sub-sample for reliability. In addition, the relationship between items also analyzed.
Thank you. In this case, results could be interesting to be considered on training and competitive games in soccer players.
The work is important because having a valid and reliable instrument
Thank you, we thought the same.
Reviewer 4 Report
Competition Seriousness and Competition Level Modulate Testosterone and Cortisol Responses in Soccer Players
Manuel Jiménez et al present a study about changes in testosterone and cortisol depending of competition seriousness and level.
The study is interesting, well conducted and contains enough information which is in agreement with other studies about the same subject. It is an important piece of information in that field of study.
Author Response
Competition Seriousness and Competition Level Modulate Testosterone and Cortisol Responses in Soccer Players
Manuel Jiménez et al present a study about changes in testosterone and cortisol depending of competition seriousness and level.
Thank you.
The study is interesting, well conducted and contains enough information which is in agreement with other studies about the same subject. It is an important piece of information in that field of study.
Thank you, we consider the same.
Round 2
Reviewer 2 Report
The authors did not take my criticisms into much consideration and according to the other referees comments, they might be right.